# Antibiotic Resistance in Wastewater Treatment Plants and Transmission Risks for Employees and Residents: The Concept of the AWARE Study

**DOI:** 10.3390/antibiotics10050478

**Published:** 2021-04-21

**Authors:** Laura Wengenroth, Fanny Berglund, Hetty Blaak, Mariana Carmen Chifiriuc, Carl-Fredrik Flach, Gratiela Gradisteanu Pircalabioru, D. G. Joakim Larsson, Luminita Marutescu, Mark W. J. van Passel, Marcela Popa, Katja Radon, Ana Maria de Roda Husman, Daloha Rodríguez-Molina, Tobias Weinmann, Andreas Wieser, Heike Schmitt

**Affiliations:** 1Institute and Clinic for Occupational, Social and Environmental Medicine, University Hospital, LMU (Ludwig-Maximilians-Universität) Munich, 80336 Munich, Germany; katja.radon@med.uni-muenchen.de (K.R.); Daloha.Rodriguez_Molina@med.uni-muenchen.de (D.R.-M.); Tobias.Weinmann@med.uni-muenchen.de (T.W.); 2Centre for Antibiotic Resistance Research at University of Gothenburg, Department of Infectious Diseases, Institute of Biomedicine, University of Gothenburg, 405 30 Gothenburg, Sweden; fanny.berglund@gu.se (F.B.); carl-fredrik.flach@microbio.gu.se (C.-F.F.); joakim.larsson@fysiologi.gu.se (D.G.J.L.); 3Centre Infectious Disease Control, National Institute for Public Health and the Environment (RIVM), 3721 MA Bilthoven, The Netherlands; hetty.blaak@rivm.nl (H.B.); mw.v.passel@minvws.nl (M.W.J.v.P.); ana.maria.de.roda.husman@rivm.nl (A.M.d.R.H.); heike.schmitt@rivm.nl (H.S.); 4Earth, Environment and Life Sciences Division, Research Institute, University of Bucharest, 050657 Bucharest, Romania; carmen.chifiriuc@bio.unibuc.ro (M.C.C.); gratiela.gradisteanu@icub.unibuc.ro (G.G.P.); luminita.marutescu@bio.unibuc.ro (L.M.); marcela.popa@bio.unibuc.ro (M.P.); 5Directorate of International Affairs, Ministry of Health, Welfare and Sport, 2500 EJ The Hague, The Netherlands; 6Institute for Medical Information Processing, Biometry, and Epidemiology—IBE, LMU (Ludwig-Maximilians-Universität) Munich, 81377 Munich, Germany; 7Pettenkofer School of Public Health, 81377 Munich, Germany; 8Division of Infectious Diseases and Tropical Medicine, LMU (Ludwig-Maximilians-Universität) University Hospital, 80802 Munich, Germany; wieser@mvp.uni-muenchen.de; 9Faculty of Medicine, Max von Pettenkofer Institute, LMU (Ludwig-Maximilians-Universität), 80336 Munich, Germany

**Keywords:** wastewater treatment plants, ESBL-producing *E. coli*, carbapenemase-producing *Enterobacteriaceae*, antibiotic resistance, employees, residents

## Abstract

Antibiotic resistance has become a serious global health threat. Wastewater treatment plants may become unintentional collection points for bacteria resistant to antimicrobials. Little is known about the transmission of antibiotic resistance from wastewater treatment plants to humans, most importantly to wastewater treatment plant workers and residents living in the vicinity. We aim to deliver precise information about the methods used in the AWARE (Antibiotic Resistance in Wastewater: Transmission Risks for Employees and Residents around Wastewater Treatment Plants) study. Within the AWARE study, we gathered data on the prevalence of two antibiotic resistance phenotypes, ESBL-producing *E. coli* and carbapenemase-producing *Enterobacteriaceae*, as well as on their corresponding antibiotic resistance genes isolated from air, water, and sewage samples taken from inside and outside of different wastewater treatment plants in Germany, the Netherlands, and Romania. Additionally, we analysed stool samples of wastewater treatment plant workers, nearby residents, and members of a comparison group living ≥1000 m away from the closest WWTP. To our knowledge, this is the first study investigating the potential spread of ESBL-producing *E. coli*, carbapenemase-producing *Enterobacteriaceae*, and antibiotic resistance genes from WWTPs to workers, the environment, and nearby residents. Quantifying the contribution of different wastewater treatment processes to the removal efficiency of ESBL-producing *E. coli*, carbapenemase-producing *Enterobacteriaceae*, and antibiotic resistance genes will provide us with evidence-based support for possible mitigation strategies.

## 1. Introduction

Antibiotic resistance has become a serious global health threat. As bacteria and certain genetic traits often move between humans, animals, and the environment, a one health approach that considers these interactions is needed to efficiently address this growing problem. The role of the environment in the emergence and dissemination of antibiotic resistance has become more and more acknowledged [1,2,3]. Still, little is known about the transmission dynamics of antibiotic-resistance determinants from water, air, and soil and their risks for humans in direct contact with these matrices [4]. A key to determining human health impacts lies in the application of epidemiological investigations, in which the carriage of antibiotic resistant bacteria (ARB) in people exposed to a specific transmission route is tested in comparison to unexposed or less exposed controls. Such studies have been carried out in travellers [5] and in agricultural settings [6,7], but other environmental exposure routes, such as via water, have rarely been studied [8,9,10,11,12].

Wastewaters from agriculture, industry, hospitals, and households are collected together at wastewater treatment plants (WWTPs), making them unintentional collection points for antimicrobials and ARB. Wastewater typically harbours a mix of residual antibiotics and other agents that are known to co-select for antibiotic resistance [13,14], which provides opportunities for selection of ARBs and hence risks for evolution and transmission of resistance. Selection pressures, together with a high density and diversity of pathogens and environmental bacteria carrying various antibiotic resistance factors, provide a milieu where new forms of resistance may emerge [15,16]. From mining of metagenomics data, we know that emergence of new antibiotic resistance genes (ARG) occurs [17,18]. Additionally, resistant bacteria already present in human faeces can pass WWTPs. For example, ESBL-producing *E. coli* (ESBL-EC) have been detected in the influent and effluent of WWTPs and the receiving surface waters [19]. It is known that human infections with ESBL-EC or carbapenem-resistant *Enterobacteriaceae* (CPE) are associated with increased mortality rates, time to effective therapy, length of hospital stay, and overall healthcare costs [20].

WWTPs are in general not developed to remove either of these (or any) resistant bacteria. Studies indicate that even though a significant reduction occurs through various treatment processes [21], significant amounts of antimicrobials, ARB, and ARGs are still shed into environmental reservoirs, including rivers and recreational water [22]. While the efficiency of conventional treatment technologies greatly differs between types of WWTPs, the role of specific treatment technologies in removal of antimicrobials, ARB, and ARGs remains poorly described [23,24]. 

Workers at WWTPs are potentially exposed to wastewaters carrying ARB and ARGs and aerosolised ARB and ARGs through different transmission routes: inhalation, dermal contact, and ingestion. Airborne bacteria have indeed been detected in WWTPs [25,26,27], including *Enterobacteriaceae* and faecal coliforms [28,29], and an increased prevalence of gastrointestinal and respiratory diseases was reported in WWTP workers, suspected to be linked to microbial exposures [30]. Although few studies so far addressed specific pathogens in WWTP workers, one has found an elevated carriage of *Tropheryma whipplei* [31]. Additionally, a higher seroprevalence of IgG against *Helicobacter pylori* was observed among sewage workers [32]. Hepatitis A virus, hepatitis E virus and positive stool PCR tests for *Leptospira spirochete* [33] were also described. However, the carriage of ARB and ARGs in WWTP workers is yet unknown. 

Furthermore, WWTPs are often located in urban settings in close proximity to residents. As bacteria can be traced back up to 150 m away from animal farms [34], neighbouring residents might also face a risk of exposure to aerosolized wastewater. WWTPs, their workers, and nearby residents therefore could represent an ideal—but yet unstudied—test case to investigate whether transmission via (waste) water actually impacts ARB and ARGs carriage. 

Within the AWARE study (Antibiotic Resistance in Wastewater: Transmission Risks for Employees and Residents around Wastewater Treatment Plants), we gather data on two antibiotic resistance phenotypes, i.e., ESBL-producing *E. coli* (ESBL-EC) and carbapenem-resistant *Enterobacteriaceae* (CPE) and on ARG prevalence from analysis of air, water, sewage, and stool samples taken from inside and outside of different WWTPs in Germany, the Netherlands, and Romania. The AWARE study specifically aims:To study carriage rates of ESBL-EC, CPE, and of a range of clinically relevant ARGs in WWTP workers and nearby residents (living within ≤300 m vicinity of a WWTP) compared to a comparison group (living 1000 m away from the closest WWTP);To study waterborne and airborne exposure to ESBL-EC, CPE, and of a range of clinically relevant ARGs in WWTP workers through ingestion and inhalation;To assess the efficiency of different WWTP treatment technologies in diminishing ESBL-EC, CPE, and a range of clinically relevant ARGs; andTo investigate selection and emergence of ESBL-EC, CPE, and a range of clinically relevant ARGs in WWTPs through studying relative changes in resistance genes and exploring putative novel resistance genes from metagenomics data.

Our overall aim with this methodological publication is to deliver precise information about the methods used in the AWARE project, including selection of participants, sample taking, creation of the questionnaire, and pilot study. This publication is a study protocol which is purely methodological and does not include results of the study. Further, we will discuss possible strengths and limitations of our study design.

## 2. Materials and Methods

### 2.1. Study Design

The AWARE study is a multicentre, cross-sectional study investigating the prevalence of ESBL-EC, CPE, and ARGs in WWTP workers, residents living within ≤300 m vicinity of a WWTP (residents), and a comparison group living >1000 m away from the closest WWTP (comparison group). The field phase is carried out in Germany (DE), the Netherlands (NL), and Romania (RO) (Figure 1.).

### 2.2. Study Population

We aim to include 450 WWTP workers (150 per country). In order to compare carriage of ESBL-EC, CPE, and ARGs, we aim to include 800 nearby residents (400 in DE, 400 in RO) living in <300 m vicinity of a WWTP (residents). Further, we aim to include 1200 residents (400 in DE, 400 in RO, 400 in NL) living >1000 m away from the closest WWTP (comparison group). Assuming an average ESBL-EC prevalence of 8% in the general population, this would allow us to detect a minimum odds ratio (OR) of 1.7 with power 80% in workers and nearby residents on a 5% significance level. 

In order to be included in the study, participants have to be within the age range of 16 to 67 years. All participants who have worked at a slaughterhouse or a farm during 12 months prior to study are excluded because contact with farm animals and working at slaughterhouses can be risk factors for ESBL-EC carriage [35]. 

### 2.3. Recruitment Process

The recruitment process for WWTP workers, residents living within ≤300 m of a WWTP, and the comparison group consisting of residents living >1000 m away from the closest WWTP is underlying local regulations and thus differs between DE, NL, and RO (Table 1). However, to control for seasonal variation of ESBL-EC, CPE, and ARGs, we aim to take all samples (water, air, stool) from the surroundings of each WWTP within eight weeks.

### 2.4. Pilot Study

We test the study methods in a pilot phase which includes recruitment of study participants, the study questionnaire, and sample taking (water, air, stool). The study questionnaire is tested by 33 participants. Fifteen participants hand in stool samples for the pilot study. Additionally, all six water and sludge samples are taken from two WWTPs.

### 2.5. Study Instruments

#### 2.5.1. Study Questionnaires

WWTP workers, residents, and members of the comparison group willing to participate receive access to an online questionnaire. However, we offer paper questionnaires at the preference of the participants. For quality control, we do double data entry with error check. The questionnaire assesses socio-demographics (age, gender, education) as well as potential risk factors for ESBL-EC, CPE, or ARGs carriage (work history, travel abroad, contact with farm animals, hospital visits, antibiotic intake, self-evaluation of general health condition). Additionally, WWTP workers also answer questions considering their specific work tasks at the WWTP, the use of personal protective equipment, and hygienic behaviour. WWTP operators answer questions about the capacity of their WWTP, origin of treated wastewaters, and wastewater treatment methods. 

Whenever possible, we retrieve questions from validated questionnaires [36,37,38,39,40,41,42,43,44,45]. Only if we cannot find validated questions, we take items from existing, but not validated, questionnaires after checking for their face validity. If we cannot find any suitable questions from previous studies, we create expert validated new items. We translate the original questionnaires from English (Appendix A) to German, Dutch, and Romanian. At least two experts on the topic who are also native speakers of the target language check the translation and provide feedback. This pre-pilot phase of the study is an iterative process to translate, back-translate, ask for feedback, and improve the current version of the questionnaires. We then test the translated questionnaires in a two-phase procedure: in the first phase, we recruit a small number of participants (*n* = 3) to read and provide verbal feedback on their understanding of each question. As we offer the questionnaire online, we create an online survey using LimeSurvey [46]. In the second phase, three persons of the target group go through the process of filling out the questionnaires online. They also provide feedback on the understanding of each question, and the online survey’s functionality. Once the questionnaire is refined and tested for clarity and understandability, it is tested in the pilot study. During the pilot study, seven WWTP operators (one from DE, six from RO) and twelve WWTP workers (three from DE, nine from RO), two nearby residents, and twelve members of the comparison group fill in the questionnaire and provide feedback. Based on the results, we refine the questionnaire. 

#### 2.5.2. Stool Samples

In DE and RO, participants receive a stool sample kit by postal service (residents and members of the comparison group) or at work (WWTP workers) after handing in an informed consent and completing the questionnaire. In NL, participants first hand in their stool samples and then fill in the questionnaire. We provide all necessary material to the participants in order to take the stool sample. This includes a paper faeces collection device, a sterile stool sampling tube, and written and drawn instructions. In DE, participants are asked to bring the stool sample directly to the next WWTP, where it is cooled or stored temporarily in a refrigerator until the next morning, when it is collected by a member of the study team. In NL, we ask participants working at a WWTP to bring their stool sample to the WWTP, where it is cooled, while residents are asked to bring it to a specified general practitioner (GP). GPs within a 2–5 km distance from selected WWTPs are approached for cooperation, to function as a collection and preservation point of stool samples. Addresses of within a 500 m radius of GPs are identified using Geographical Information System (GIS) software (version ArcGis 10.6.1). Participants who are unable to bring their sample to the GP at the indicated time/day are given the opportunity to send the samples per mail without cooling (although samples shipped per mail will be excluded from metagenomic sequencing). In RO, we ask participants to cool the stool samples at 1–10 °C directly after samples were taken and to bring them to the WWTP the next day. The same day, the stool samples are transported to the laboratory and processed within 72 h. We tested this procedure in the pilot study with fifteen participants (one WWTP operator, three WWTP workers, and eleven members of the comparison group).

At the local laboratories in DE, NL, and RO, all stool samples are inoculated directly onto the following agars: TBX or MacConkey, ChromID ESBL, ChromID OXA-48, and ChromID CARBA and incubated at 36 ± 2 °C for 24–48 h. In case of positive results, a total of two isolates belonging to the ESBL-EC phenotype and 5 isolates belonging to CPE phenotype are collected, screened for antibiotic resistance and identified by MALDI-TOF MS (Matrix Assisted Laser Desorption Ionization-Time of Flight Mass Spectrometry). We then process stool samples for DNA isolation after intermediate storage at −80 °C, which we then will use for subsequent metagenomics and qPCR analyses.

#### 2.5.3. Water Samples

We collect water samples from WWTPs at four different treatment stages: wastewater influent (WI), effluent (WE), liquid sludge from the main biological reactor (e.g., aeriation tank) (AT), and dewatered sewage sludge after thickening (S). We also take water samples from the receiving surface water 200 m upstream (WU) and 200 m downstream (WD) of the WWTP. The following Figure 2 provides an overview of the collection points of water sample, as well as stool and air samples taken. We tested this procedure in the pilot study at one WWTP in DE and two in RO.

We collect upstream (WU) and downstream (WD) water samples as close as possible to the WWTP to minimize the influence of other sources, but at enough distance to minimize the chance of diffusion to upstream locations and to ensure sufficient mixing with effluent for downstream locations. If accessible, we choose locations at 200 m upstream and 200 m downstream for waters with a width <20 m, according to the rule of thumb that complete mixing occurs at a distance of at least 10× the width of the surface water. Additionally, we choose the upstream and downstream locations in a way that no additional side streams enter the river between these locations and the effluent discharge point. Therefore, we choose locations closer to the WWTP when side streams are present within the optimal distance. We take subsurface samples according to international guidelines (ISO 19458:2007: Water quality—Sampling for microbiological analysis).

The sampling points for wastewater influent (WI) and effluent (WE) are determined by the location of the flow-proportional auto samplers at the individual WWTPs, when present. Influent samplers are usually located directly after mechanical treatment and effluent samplers after completion of treatment, prior to discharge. Using auto samplers, experienced WWTP or laboratory staffs collect 24-h flow proportional samples, of which 1 L is transferred to a sterile bottle at the end of the usual time interval applied in the WWTP (e.g., 9:00 in the morning). If no automatic samplers are available, we take grab samples from wastewater influent and effluent, at approximately 40 to 60 percent of the water depth, at a site with maximal turbulence to ensure good mixing and the possibility of solids settling is minimized. The most desirable sampling locations for grab samples of influent include: (a) the upflow siphon following a comminutor (in absence of grit chamber); (b) the upflow distribution box following pumping from main plant wet well; (c) aerated grit chamber; (d) flume throat; (e) pump wet well when the pump is operating; or (f) downstream of preliminary screening. 

When possible, we take influent samples upstream from side stream returns. We collect grab samples of effluent at the site specified in the sampling plan, or if no site is specified, we select the most representative site downstream from all entering wastewater streams prior to discharge into the receiving waters.

We take the liquid sludge sample (AT) from the main biological reactor (e.g., aeration tank). The selection of the sampling points depends on (a) the practicality of interrupting safely a stream of moving liquid sludge or cake when manually sampling; and (b) the nature of the chamber or tank design with respect to stratification of liquid sludges.

We take the sample of dewatered sewage sludge after thickening (S). Prior to the proposed sampling date, we assess sludge processing (dewatering and treatment) to ensure that sludge is in the appropriate form (liquid versus dewatered, untreated cake versus treated biosolids) and is available for sampling at the proposed date, time, and sampling point. If needed, we will adjust the selection points.

After all water and sludge samples are collected, they are kept at 1–10 °C at the WWTP and transported at 1–10 °C to the laboratory in NL (samples from DE and NL) and RO (samples from RO). At the laboratories in NL and RO, we process all samples within 48–72 h after sampling, e.g., homogenization (for sludge) and membrane filtration (for sludge and water). We then process water filters for DNA isolation, which we use for subsequent metagenomics and qPCR analyses.

#### 2.5.4. Air Samples

We intend to ask a subset of 50 workers from 10 WWTPs per participating country to collect air samples to analyse personal exposure. Sampling is based on GSP inhalable sampling heads equipped with Teflon filters on Gilair pumps (3.5 L/min), sampling the total inhalable air of workers whose job position included activities at different treatment stages.

The pumps are programmed and fixed at the worker’s belt or pocket by a member of the study team. A study team member checks the correct functioning of the pumps at the beginning, after three hours, and after six hours of sampling. After six hours, the study team member turns off the pumps. We wrap the heads of the pumps in aluminium foil and transport them directly to the laboratory where the pumps are opened on a sterile work bench. The laboratory assistants remove the Teflon filters with a pair of sterile tweezers and freeze them at −20 °C (DE) or −80 °C, respectively (NL, RO). We ship all filters to NL for analysis. Feasibility of the procedures is checked during the pilot study.

### 2.6. Metagenomic Analysis

The Swedish and Romanian team conduct culture-independent analyses. They will employ shotgun metagenomics sequencing [47,48,49] by the Illumina NovaSeq technology. This enables simultaneous quantification of any known antibiotic-resistance gene if present at sufficiently high levels to allow detection. In addition, shotgun metagenomics allows for the analysis of mobile genetic elements such as integrons and transposons and of the taxonomic composition of the microbial communities [49]. Although costs for DNA sequencing have dropped dramatically, it still involves substantial costs if relatively rare resistance genes are targeted in complex community samples [48,50]. Therefore, we will select a subset of air, sewage, water, and faecal samples for sequencing, while we plan to choose 24 genes for qPCR investigations in all human, water, and air samples. The selection will be based on an initial screen using qPCR arrays with considerably more genes for a subset of samples. Antibiotic residues and their metabolites are usually detected in the environment at trace levels but may still be present at concentrations that have the potential to select for microbial resistance [49,51] and possibly also induce horizontal gene transfer [52]. Therefore, residues are monitored by high-performance liquid chromatography interfaced with tandem mass spectrometry (HPLC–MS/MS) in selected plants, including the WWTPs in which metagenomics data are also determined. We perform sample selection for metagenomic analyses by using propensity score matching of the exposed and unexposed groups to achieve proportional and non-statistically significant balance of the groups at a 5% statistical level. 

### 2.7. Data Management

We store the personal contact data of participants and the history of contacts via letters, e-mails, and phone calls in a password protected Access database separated from questionnaire and sample data. We pseudonymize all assessed data. The laboratories document results of stool, air, and water samples in Excel. We primarily do data cleaning and analysis in R. Additional software will be used depending on the specific analyses. All personal data are stored password protected with access only to the members of the study team. We ensure that data management is bound to FAIR principles [53], e.g., including storage of research data obtained in publicly accessible and findable repositories.

### 2.8. Statistical Analysis

For descriptive analyses we assess the distribution of numerical variables visually for normality using histograms and present the mean ± standard deviation if normally distributed or the median ± inter-quartile range if non-normally distributed. We present categorical variables using absolute and relative frequencies. We handle missing values by multiple imputation in case of missing at random or missing completely at random. We do data cleaning, as well as multiple imputation, propensity score matching, data presentation, and outcome models using the statistical software R version 3.5 and up [54]. Additional software will be used and documented depending on the specific analyses. 

We perform bivariate hypothesis testing choosing an appropriate statistical test depending on the type of variables involved, their distribution, and the number of counts per cell (for categorical variables). We perform logistic crude and adjusted regression models for the main outcomes such as carriage of ESBL-EC, CPE, and ARGs. Main exposure variables will include whether a participant belongs to the group of WWTP workers, nearby residents, or the comparison group. We consider linear regression models for secondary outcomes if these are numerical. We present results from regression models with the point estimate and its corresponding 95% confidence interval. We do variable selection for the models using a combination of experts’ opinion from within the AWARE consortium, evidence in the current literature, and the use of Directed Acyclic Graphs (DAGs).

## 3. Discussion

To our knowledge, this is the first study investigating the potential spreading of ESBL-EC, CPE, and ARGs from WWTP to workers, the environment, and nearby residents. By involving different European countries, covering a variety of different types of WWTPs, our results will be relevant for a large number of situations. The methodological combination of epidemiology, molecular biology, and metagenomics will allow us to draw multilevel conclusions. We demonstrated feasibility of the AWARE project in the pilot study.

Our study is carried out cross-sectionally at each WWTP. Thus, the study does not provide information how the numbers of ESBL-EC, CPE, and ARGs vary with time/seasons. It is possible that bias arises for some samples due to different laboratories analysing them. In order to minimize such biases, we develop all SOPs jointly and centralize sample preparation and analyses whenever possible. WWTP workers are organized in different ways depending on the country: In NL, WWTP workers do not work at one specific WWTP, hampering the comparison between ESBL-EC, CPE, and ARGs at the selected WWTP and in stool from workers. 

Our assessment of transmission of antibiotic-resistant bacteria from WWTPs to the surrounding environment will enable us to formulate recommendations, such as adapted sewage treatment, or recommendations for a minimal distance between WWTPs and residential buildings in order to reduce transmission of antibiotic resistant bacteria. 

## Figures and Tables

**Figure 1 antibiotics-10-00478-f001:**
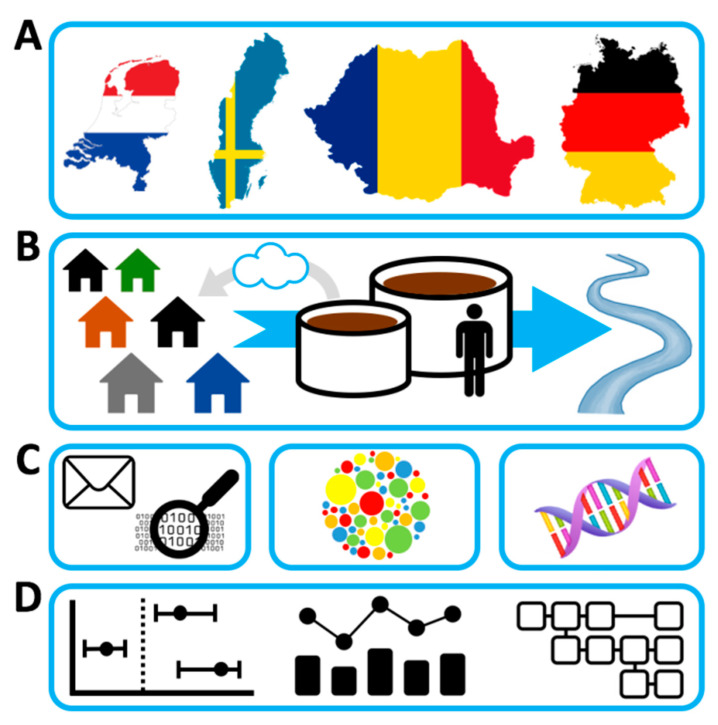
Legend Graphical Abstract: Overview of AWARE study, with (**A**) the participating countries, (**B**) the study domain, wastewater treatment plant samples, and workers of and residents living nearby wastewater treatment plants, (**C**) the techniques involved (questionnaire, molecular and cultural analyses of ESBL-EC, CPE, and the resistome, and (**D**) the outcome: epidemiological evaluation of differences in prevalence of ESBL-EC, CPE, and the resistome between workers and residents of wastewater treatment plants and the general population, changes in relative and absolute resistance along different wastewater chains, and models for airborne and waterborne exposure to resistant bacteria and resistance genes.

**Figure 2 antibiotics-10-00478-f002:**
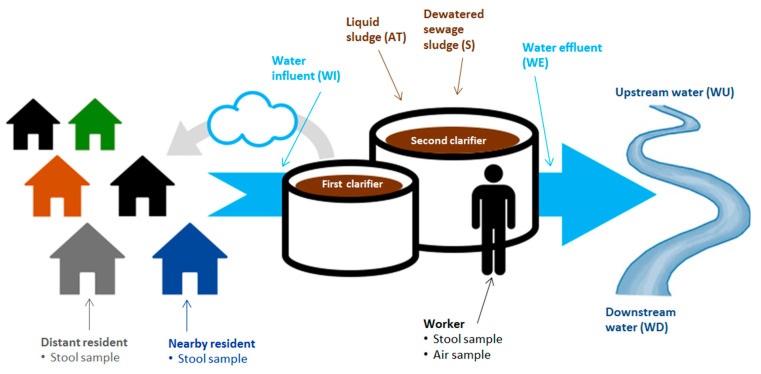
Collection points of water, air, and stool samples.

**Table 1 antibiotics-10-00478-t001:** Recruitment of participants into the AWARE study.

	Germany	The Netherlands	Romania
Selection of WWTPs	Eligible WWTPs are selected due to the following criteria: There are residents living in <300 m vicinity of WWTP, WWTP is located close enough to laboratories for the analyses of samples	All 21 regional waterboards ^3^ are included.	WWTPs are chosen to assure a good representativeness of different regions across the country.
Invitation of WWTPs	The operators of the WWTPs are contacted by the local study team and asked to participate.	The waterboards are informed of the study through the Dutch Water Authorities and asked to participate.	The operators of the WWTPs are contacted by the local study team and asked to participate.
Response in WWTPs	8 WWTPs are interested in participating.	12 waterboards are interested in participating ^4^.	9 WWTPs are interested in participating.
Study presentation and informing of WWTP workers	The study team visits 6 interested WWTPs and presents the project to the workers ^1^.	The WWTP workers of 10 waterboards are invited to attend a presentation of the study by the local study team ^5^. The workers of the remaining 2 waterboards are recruited internally through email. By sending the presentation to all workers via email, also workers not attending the meeting are reached.	The WWTP operators inform and invite the employees to participate. Afterwards, several short information sessions are organized at the WWTPs for recruiting participants.
Informing of nearby residents	The study team researches the street names of all streets within ≤300 m vicinity of a participating WWTP through Google Maps and asks the local registration office ^2^ for the full address of all persons aged 16–67 years and having their main residence in those streets.	Due to concerns of the waterboards, residents living in ≤300 m vicinity of a WWTP cannot be included.	Invitations to the study are done using door-to-door approach. Additionally, in public places like streets, parks, and markets, potential participants are orally addressed and information sheets with details about the study are distributed. The participants are at least 18 years old.
Informing of comparison group	The addresses are collected in the same way as for the nearby residents, except that addresses >1000 m away from the closest WWTP and close to a train station are chosen to allow fast transportation of samples by the study team.	All addresses within a 500 m radius of GPs, who are willing to cooperate, are identified ^6^. Then, 300–500 addresses per GP are randomly selected to extract personal data from the Dutch Personal Records Database (BRP). Information on the study is sent to all residents living at the selected addresses over 16 years of age.	Same procedure as for nearby residents
Incentives for participants ^7^	Participants participate in a raffle with 10 shopping vouchers with a total value of 1500 Euros.	Every participant receives a gift card worth 20 Euro.	Every participant is granted 5 Euro.
Timing of sample taking	To control for seasonal variation of ESBL-EC, CPE, and ARGs all samples (water, air, stool) from the surroundings of one WWTP are aimed to be taken within eight weeks.		

^1^ Two WWTPs stepped back from participation because they feared that residents and media might complain about WWTPs in case ESBL-EC, CPE, or ARGs would be found in their WWTP. ^2^ If addresses cannot be retrieved from the local registration offices, members of the study team go from door to door to recruit participants. In case of no reply, up to two reminders are sent (7 and 21 days after initial invitation). Further methods will be performed to increase the response: newspaper articles describing the AWARE project published by local newspapers, online advertisement on the study’s Facebook page and in groups like notice boards and job advertisements, flyers about the AWARE study in doctors’ offices of local physicians, invitations via e-mail to workers from different work fields (industry and public sector). ^3^ Waterboards are regional government bodies supervising, e.g., sewage treatment in their respective regions. ^4^ Nine waterboards did not want to participate out of fear for causing commotion among nearby residents or workers, or lack of interest to invest time and/or manpower to help organize recruitment. ^5^ WWTP workers generally work at multiple WWTPs, making it impossible to study workers of specific WWTPs. Therefore, all workers of waterboards were invited to participate, but only a selection of WWTPs (1–3 per waterboard) are selected for environmental sampling. ^6^ General practitioners (GP) within a 2–5 km distance from selected WWTPs are approached for cooperation, to function as a collection and preservation point of stool samples. Addresses of within a 500 m radius of GPs are identified using Geographical Information System (GIS) software (version ArcGis 10.6.1). ^7^ Participants who hand in a stool sample and a completed questionnaire.

## Data Availability

Data will be available via www.aware-study.eu, once data cleaning and analyses have been completed.

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
