# Peer review of "Antibiotic Resistance in Wastewater Treatment Plants and Transmission Risks for Employees and Residents: The Concept of the AWARE Study"

_antibiotics, 2021, doi:10.3390/antibiotics10050478_

Round 1

Reviewer 1 Report

- The title of the manuscript sounds informative and attractive.

- Abstract. Please be careful in the abstract with the usage of abbreviations. Their meaning might seem obvious but they may cause plenty of confusion. So please avoid the usage of abbreviation in the abstract. Could you please more highlight the uniqueness of your study here to make the abstract more attractive? Usually, the abstracts include the obtained results, please also take this into account.

- Introduction that serves here as a theoretical background of the study is nicely developed but lacks, in my opinion, depth as is rather descriptive. Please expand here. What could be useful is adding a table where the recent key results are summarized. I think the first sentence (lines 43-44) is redundant.

- I think that authors should be clearer about the aim of the study right in the Introduction. There are some places where the aim of the study is indicated but the readers should be aware of what to expect sooner. Please located to the end of the Introduction a short summary of sections that follow. Please be careful with abbreviations in this part to avoid any confusion.

- The methodology of the experiment seems to be nicely developed. It is very extensive, which makes it difficult for the reader to understand the topic. I suggest placing diagrams, tables to increase the clarity of the methodology (number of trials, etc.). Figure 1 and Table 1 are not enough, please expand this part with some graphic elements.

- The assumptions of the methodology at this stage seem to be correct.

After the methodological part, the authors go straight to the discussion. The authors emphasize the lack of research on the potential spread of ESBL-EC, CPE and ARGs from WWTP to workers, the environment and nearby residents. However, part of the discussion should be expanded to include reports from partial studies.

I understand that there is no such broad approach to the problem so far, but you can use the results with a smaller scope. You should expand on this part of the study. Lists of results in a table or in another form are welcome.

- Is it please possible to develop (based on literature retrieval) a couple of hypotheses that were tested by the experiment. I think that this would be useful for better understanding what was expected to be achieved and what was really achieved.

- The list of references should be expanded to better cover the recent debate.

- Let me thank authors for the work on the manuscript they did so far. I think that the manuscript will be a good fit for Antibiotics but some work still has to be done. In the current version, the manuscript is not ready for publishing and a major revision has to be done. I hope that the authors find my comments useful. I´m honoured having an opportunity to review the manuscript for the journal.

Reviewer 2 Report

The study is relevant within the field of environmental microbiology as antibiotic resistant bacteria/genes (ARB, ARGs) are a current threat to public health and staffs working in wastewater treatment plants are exposed to many hazards. The proposed design to test the hypothesis of whether neighboring residents of WWTPs are at increased risk of exposure to ARB; however, it was noted only people working in slaughterhouse or farm will not be included. Antibiotic resistance can be affected by food, water, sanitation, hygiene, and many more. Although it will give a rough idea that WWTPs might be the reason for the increase/decrease of ARGs/ARB it might not be enough to clearly elucidate the role of WWTPs. Further criteria for exclusion or inclusion of participants are suggested.

Line 28- Please revise word “wastewater” throughout the manuscript

Line 43- Please delete the whole line.

Line 74- Relevant papers e.g. Thakali et al, Reduction of ARGs at two conventional wastewater treatment plants in Louisiana, USA, can be cited.

Round 2

Reviewer 1 Report

Dear authors, thank you for responding to your comments. However, I didn't mean to answer the question with a question (who is the reviewer?), Because I have the impression that the answers are often evasive. The only thing missing from the study is a graphic representation of the methodology and research concept. Maybe something like a graphic abstract? Please think about it and complete the study with this element. Maybe Fig in the text should be taken into account, supplemented with entries about the type of analyzes performed and the general concept of the research. 

Author Response

We thank the reviewer for the second round of revision. We added a graphical abstract to the publication.